# ROBUSTNESS AND EQUIVARIANCE OF NEURAL NETWORKS

## ABSTRACT

Neural networks models are known to be vulnerable to geometric transformations as well as small pixel-wise perturbations of input. Convolutional Neural Networks (CNNs) are translation-equivariant but can be easily fooled using rotations and small pixel-wise perturbations. Moreover, CNNs require sufficient translations in their training data to achieve translation-invariance. Recent work by Cohen & Welling (2016), Worrall et al. (2016), Kondor & Trivedi (2018), Cohen & Welling (2017), Marcos et al. (2017), and Esteves et al. (2018) has gone beyond translations, and constructed rotation-equivariant or more general group-equivariant neural network models. In this paper, we do an extensive empirical study of various rotation-equivariant neural network models and Standard CNNs to understand how effectively they learn rotations. This includes Group-equivariant Convolutional Networks (GCNNs) by Cohen & Welling (2016), Harmonic Networks (H-Nets) by Worrall et al. (2016), Polar Transformer Networks (PTN) by Esteves et al. (2018) and Rotation equivariant vector field networks by Marcos et al. (2017). We empirically compare the ability of these networks to learn rotations efficiently in terms of their number of parameters, sample complexity, rotation augmentation used in training. We compare them against each other as well as Standard CNNs. We observe that as these neural networks learn rotations, they instead become more vulnerable to small pixel-wise adversarial attacks, e.g., Fast Gradient Sign Method (FGSM) and Projected Gradient Descent (PGD). In other words, robustness to geometric transformations in these models comes at the cost of robustness to small pixel-wise perturbations.

## 1 INTRODUCTION

Neural network-based models achieve state of the art results on several speech and visual recognition tasks but these models are known to be vulnerable to various adversarial attacks. Szegedy et al. (2013) show that small, pixel-wise changes that are almost imperceptible to the human eye can make neural networks models grossly misclassify. They find a small perturbation so as to maximizes the prediction error of a given model using box-constrained L-BFGS. Goodfellow et al. (2015) propose the Fast Gradient Sign Method (FGSM) as a faster approach to find such an adversarial perturbation given by $x' = x + \epsilon \operatorname{sign}(\nabla_x J(\theta, x, y))$, where $x$ is the input, $y$ represents the targets, $\theta$ represents the model parameters, and $J(\theta, x, y)$ is the cost used to train the network.

Subsequent work has introduced multi-step variants of FGSM, notably, an iterative method by Kurakin et al. (2017) and Projected Gradient Descent (PGD) by Madry et al. (2018). On visual tasks, the adversarial perturbation must come from a set of images that are perceptually similar to a given image. Goodfellow et al. (2015) and Madry et al. (2018) study adversarial perturbations from the $\ell_\infty$-ball around the input $x$, namely, each pixel value is perturbed by a quantity within $[-\epsilon, +\epsilon]$. Broadly, all the above-mentioned adversarial attacks are model-dependent. Tramer et al. (2017) also mention model-agnostic perturbations using the direction of the difference between the intra-class means.

There is a large class of spatial transformations including translations, rotations, scaling that preserve perceptual similarity. Convolutional Neural Networks (CNNs) are translation-equivariant by construction. Recent works show that apart from pixel-wise perturbations, geometric transforms like rotations are also possible natural attacks; see Dumont et al. (2018), Gilmer et al. (2018). En-

gstrom et al. (2017) show that simple adversarial attacks using rotations and translations can fool CNNs, even when they are adversarially trained to make them robust to $\ell_p$-bounded adversaries. They observe that $\ell_p$-bounded and spatial adversarial perturbations have additive or super-additive effect on the performance drop, suggesting that these two types of attacks have no bearing on each other. Engstrom et al. (2017) also show that CNNs achieve translation invariance only if the training data (or augmentation) contains some amount of translated inputs, however, their accuracy against the worst-case translations is significantly worse than the average-case.

CNNs are translation-equivariant but not equivariant with respect to other spatial symmetries such as rotations, reflections etc. Variants of CNNs to achieve rotation-equivariance and other symmetries have received much attention recently, notably, Harmonic Networks (H-Nets) by Worrall et al. (2016), cyclic slicing and pooling by Dieleman et al. (2016), Tranformation-Invariant Pooling (TI-Pooling) by Laptev et al. (2016), Group-equivariant Convolutional Neural Networks (GCNNs) by Cohen & Welling (2016), Steerable CNNs by Cohen & Welling (2017), Deep Rotation Equivariant Networks (DREN) by Li et al. (2017), Rotation Equivariant Vector Field Networks (RotEqNet) by Marcos et al. (2017), Polar Transformer Networks (PTN) by Esteves et al. (2018).

For our study, we do the experiments on MNIST, Fashion MNIST and CIFAR10 and choose GCNNs as they achieve close to the current state of the art results on MNIST-rot[1] and CIFAR10 data sets as reported in Esteves et al. (2018). GCNNs provide good representative networks to understand the effect of $\ell_p$-bounded and spatial transformation adversaries on symmetric networks. GCNNs use G-convolutions, they have more weight-sharing than regular convolution layers, and they are easy to implement with minimal computational overhead for discrete groups of symmetry generated by translations, reflections, and rotations. We do show similar qualitative trends on Harmonic Networks (H-Nets), Polar Transformer Networks(PTN) and Rotation Equivariant Vector Field Networks (RotEqNet).

## 2 ROBUSTNESS OF EQUIVARIANT NETWORKS

We study the robustness of StdCNNs and GCNNs to adversarial attacks based on rotations as well as pixel-wise perturbations for MNIST, Fashion MNIST and CIFAR10 data sets. To the best of our knowledge this is the first study of a rotation-equivariant network towards pixel-wise perturbations. There are other types of networks like CapsNetSabour et al. (2017) which do show natural robustness to AffNist dataset[2] though not trained on it. And further derived networks based on EM Routing Hinton et al. (2018) which along with better spatial robustness also seem to be robust to FGSM attacks. We have also checked the robustness of CapsNet to small rotations.

### 2.1 ROBUSTNESS TO ROTATIONS

We study the robustness of equivariant networks to attacks based on rotations for MNIST, Fashion MNIST and CIFAR10. The main takeaways of our empirical results are (a) Rotation equivariant networks are robust to only small degrees of rotations away from the ones present in the training data, (b) applying data augmentation increases their robustness, (c) Rotation equivariant networks do achieve state of the art results with smaller sample size for training.

We first trained all the networks on MNIST, Fashion MNIST and CIFAR10 respectively with no rotation augmentation and tested against inputs augmented with varying range of random rotations from $\pm 0°$ to $\pm 180°$. In Figure (1)(left), (2)(left) and (14)(left) we observe the inherent robustness of all the equivariant networks to small(below $\pm 40°$) rotations. Most equivariant networks (except HNets for Fashion MNIST) accuracy is always greater than StdCNNs. We also did the same experiment for CapsNet(see Figure 13) with MNIST and found their values to be sandwiched between GCNNs and HNets. Figure (1)(right), (2)(right) and (14)(right) shows the performance of these networks to the entire range of rotations. Here we observe PTN and RotEqNet to be more robust than the other networks for MNIST and Fashion MNIST. While GCNN is better than StdCNN for CIFAR10.

---

[1] http://www.iro.umontreal.ca/ lisa/twiki/bin/view.cgi/Public/MnistVariations
[2] https://www.cs.toronto.edu/ tijmen/affNIST/

In Figure 3 we observe that since MNIST contains some natural rotation augmentation in the training data, equivariant networks are able to utilize this augmentation to become more invariant to small rotations than StdCNNs. However, such augmentation is not present in Fashion MNIST and CIFAR10, so they donot perform significantly better than StdCNNs.

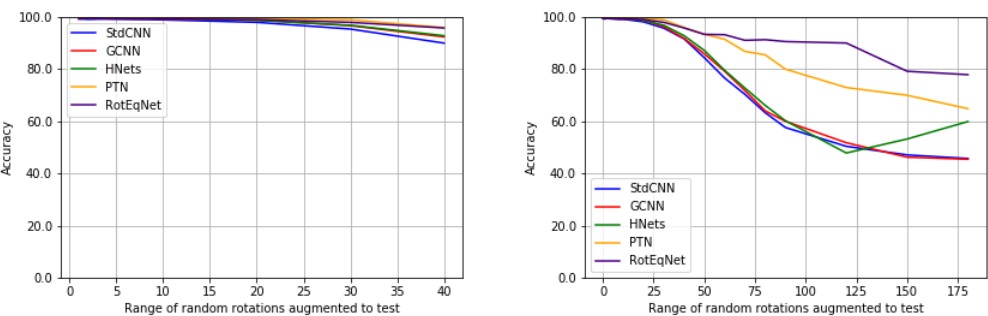

Figure 1: On MNIST, networks trained with no augmentation, test augmented with random rotations in $[-x°, x°]$ range. (left) Rotations(small) upto $40°$ (right) Rotations from $0°$ to $180°$

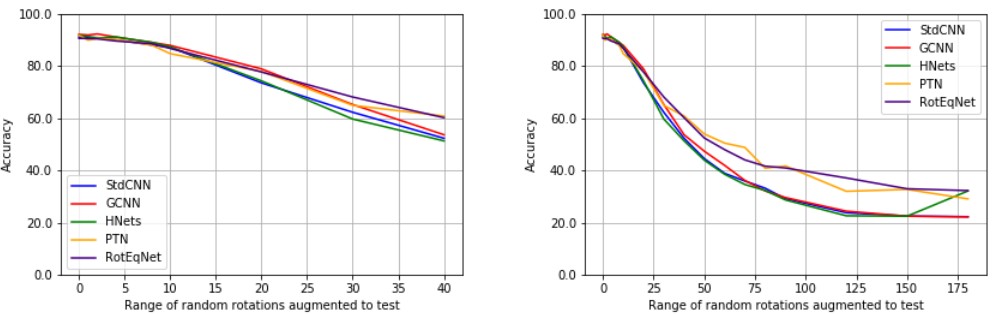

Figure 2: On Fashion MNIST, networks trained with no augmentation, test augmented with random rotations in $[-x°, x°]$ range. (left) Rotations(small) upto $40°$ (right) Rotations from $0°$ to $180°$

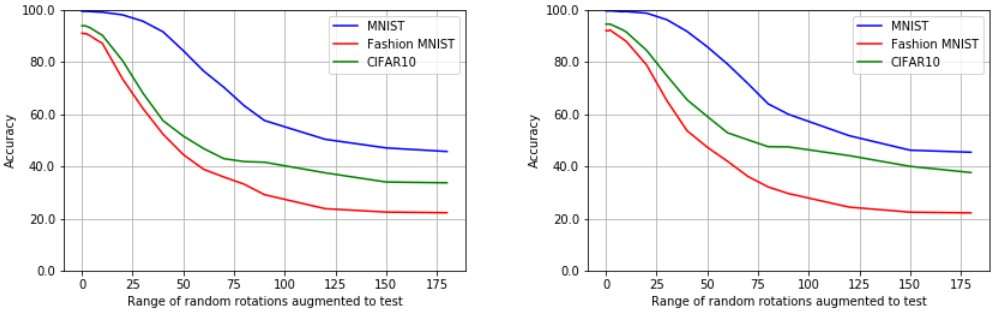

Figure 3: Robustness of (left) StdCNN and (right) GCNN to rotations without training augmentation when test augmented with random rotations in the range $[-x°, x°]$ for MNIST, Fashion MNIST and CIFAR10 datasets.

### 2.1.1 ROBUSTNESS WITH TRAINING AUGMENTATION

We trained the equivariant networks with input augmented with varying range of random rotations from $±0°$ to $±180°$. We observe that when we compare the networks accuracy when trained with

unrotated data against training with data augmented with $\pm180°$ range of random rotation augmentations only on MNIST rotation-equivariant networks achieve almost($\approx 1\%$ difference) the same performance while there is a gap of 3-5% for Fashion MNIST and CIFAR10. This is evident in Figure 4 where we see a change in slope of the black line from $\pm0°$ to $\pm180°$. The change is much more significant for StdCNNs which is evident from the change in slope of the brown line from $\pm0°$ to $\pm180°$. From this it is very clear that rotation-equivariant networks depend very heavily on augmentation to learn even small rotations. Since, even with augmentation there is a gap between the accuracy of the network against that trained without augmentation, it strongly indicates that even these networks trained with augmentation *rotations* itself can be adversarial for the network. Similar observation is made for StdCNNs by Engstrom et al. (2017).

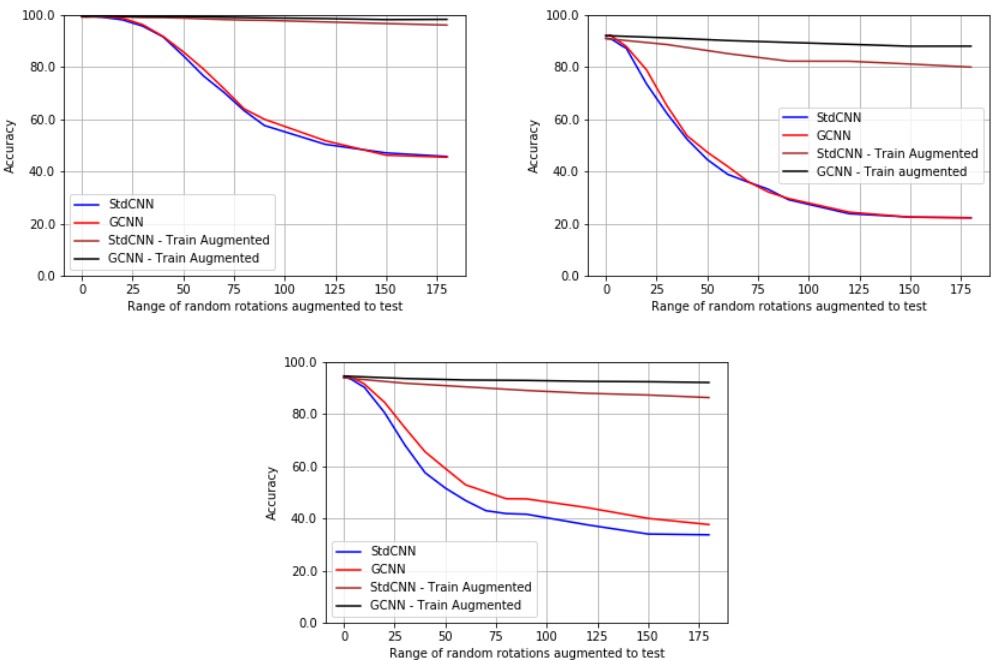

Figure 4: Networks trained with and without augmentation to dataset, random rotation augmentations in $[-x°, x°]$ range. (left) MNIST (centre) Fashion MNIST (right) CIFAR10

## 2.2 SAMPLE COMPLEXITY OF NETWORKS

To understand the sample complexity of the networks, we perform two experiments. In the first we train the networks with varying sample sizes of MNIST training set and test them on the entire MNIST test set. And in the second experiment we do the same as the first with the inputs in train and test augmented with random rotations in the range $[-180°, 180°]$. From Figure 5, we can see that rotation equivariant networks achieve their best performance safely using $10k$ - $30k$ training samples. This confirms that rotation equivariant networks do reduce training sample size.

## 2.3 ROBUSTNESS TO PIXEL-WISE PERTURBATIONS

From Figure 4 we observed that only with augmentation the networks become robust to rotations. Hence, we augment the networks with varying range of rotation from $\pm0°$ to $\pm180°$ and study in detail the vulnerability of StdCNNs and GCNNs to pixel-wise perturbations e.g. FGSM and PGD for MNIST, Fashion MNIST and CIFAR10 datasets.

For MNIST and Fashion MNIST, we give the accuracy for the networks attacked with FGSM at various values of $\epsilon$ in the range $[0, 1]$ with and without test augmentation and follow it up with a comparion at $\epsilon = 0.3$. For CIFAR10 we do the unrotated versus rotated test augmentation comparison at $\epsilon = 0.06$.

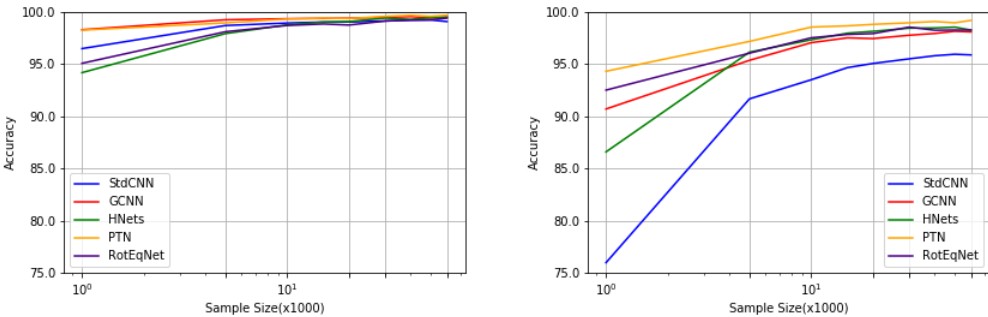

Figure 5: Networks trained with varying training sample size on X-axis. (left) Only MNIST, (right) MNIST train and test augmented with random rotations in $[-180°, 180°]$ range.

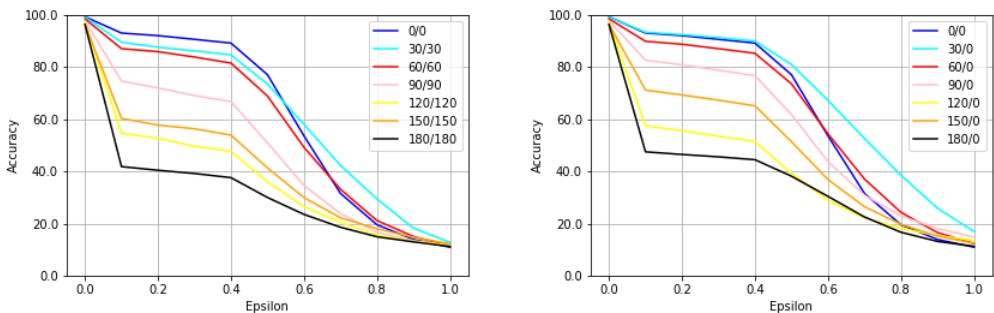

Figure 6: StdCNNs, MNIST adversarially attacked with FGSM at varying epsilon ball as perturbation budget on X-axis. (left) train and test augmented with $[-x°, x°]$ range (right) Only train augmented with $[-x°, x°]$ range and no test augmentation.

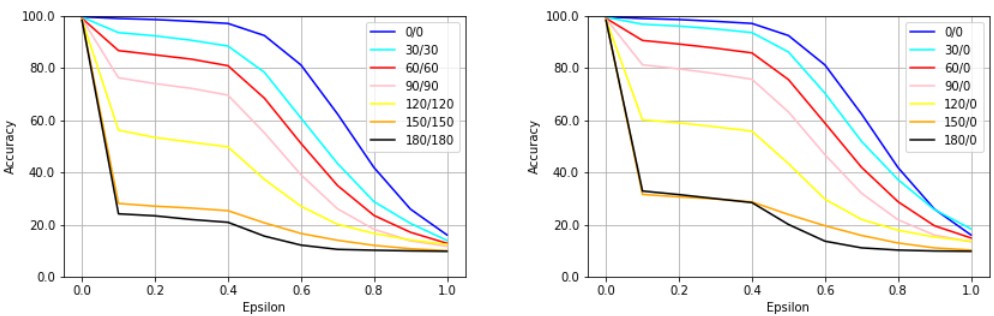

Figure 7: GCNNs, MNIST adversarially attacked with FGSM at varying epsilon ball as perturbation budget on X-axis. (left) train and test augmented with $[-x°, x°]$ range (right) Only train augmented with $[-x°, x°]$ range and no test augmentation.

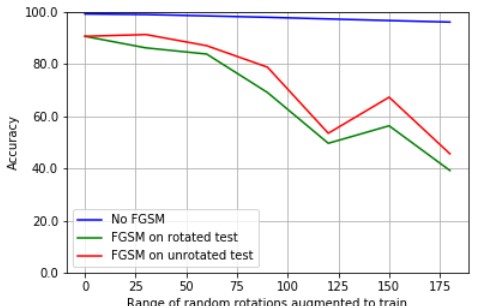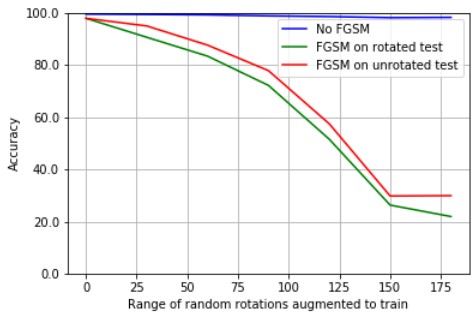

Figure 8: On MNIST, Comparison of network with/without FGSM on rotated and unrotated test with $\epsilon = 0.3$. Train/test if augmented are with random rotations in $[-x^\circ, x^\circ]$ range (left) StdCNNs, (right) GCNNs.

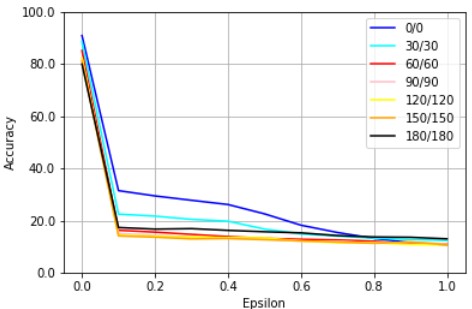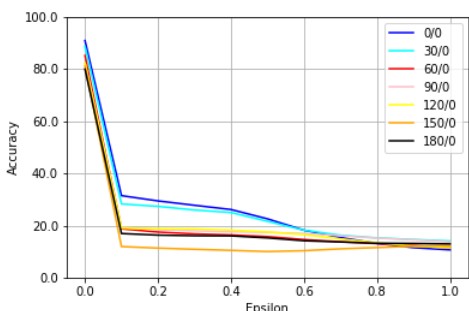

Figure 9: StdCNNs, Fashion MNIST adversarially attacked with FGSM at varying epsilon ball as perturbation budget on X-axis. (left) train and test augmented with $[-x^\circ, x^\circ]$ range (right) Only train augmented with $[-x^\circ, x^\circ]$ range and no test augmentation.

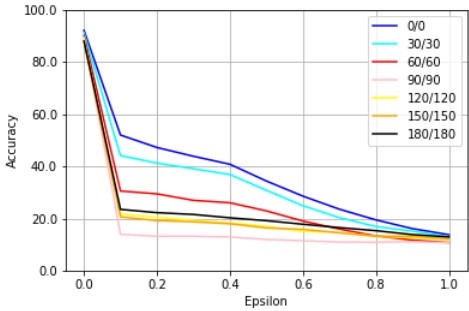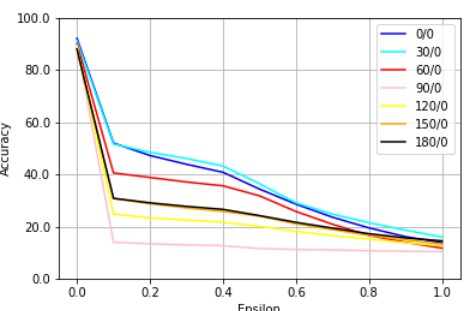

Figure 10: GCNNs, Fashion MNIST adversarially attacked with FGSM at varying epsilon ball as perturbation budget on X-axis. (left) train and test augmented with $[-x^\circ, x^\circ]$ range (right) Only train augmented with $[-x^\circ, x^\circ]$ range and no test augmentation.

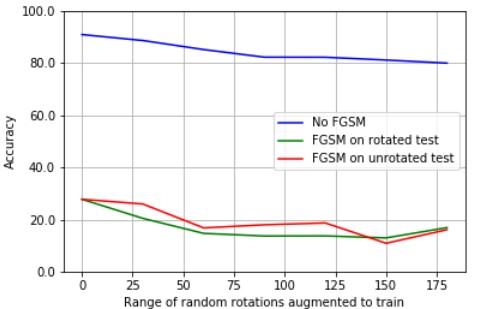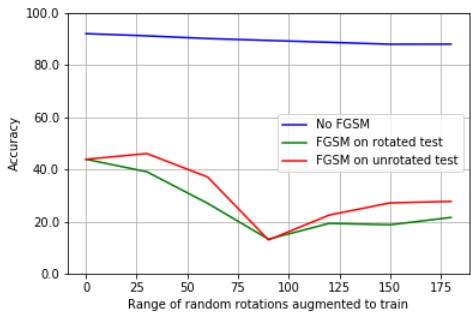

Figure 11: On Fashion MNIST, Comparison of network with/without FGSM on rotated and unrotated test with $\epsilon = 0.3$. Train/test if augmented are with random rotations in $[-x°, x°]$ range. (left) StdCNNs, (right) GCNNs.

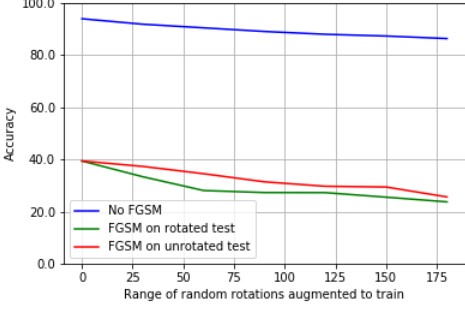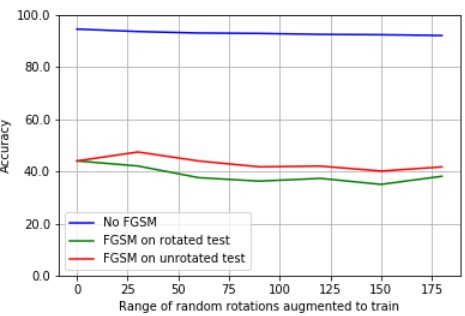

Figure 12: On CIFAR10, Comparison of network with/without FGSM on rotated and unrotated test with $\epsilon = 0.06$. Train/test if augmented are with random rotations in $[-x°, x°]$ range. (left) StdCNNs, (right) GCNNs.

*Accuracy drop with varying $\epsilon$*

It is clear from the above experiments that as GCNNs become more robust to larger rotations, they become more vulnerable to pixel-wise attacks, in comparison to StdCNNs. We do a finer analysis of the attacks with varying $\epsilon$ values, $\epsilon$ being the maximum perturbation allowed for the attack. Plots in Figure 6 and 9 are for FGSM attack on StdCNNs with changing $\epsilon$ for MNIST and Fashion MNIST, respectively and Figure 12(left) is for FGSM attack on StdCNNs with $\epsilon = 0.06$ for CIFAR10. Plots in Figure 7 and 10 are for FGSM attack on GCNNs with changing $\epsilon$ for MNIST and Fashion MNIST, respectively and 12(right) are for FGSM attack on GCNNs with $\epsilon = 0.06$ for CIFAR10. For PGD attack plots please refer to Figure 15, 18 and 21(left) for StdCNNs and 16, 19 and 21(right) for GCNNs in the appendix. The labels a/a in the legend of the plots denote the $[-a°, a°]$ range of random rotations augmented to train/test, respectively. We observe that even for $\epsilon$ as small as 0.1 the networks exhibit a behaviour similar to that seen from above experiments. *As GCNNs become robust to larger rotations they become more vulnerable to pixel-wise attacks even for smaller epsilon.*

In Figure 33, we also have checked the vulnerability of RotEqNet for MNIST to pixel-wise perturbations like FGSM and PGD. There is no clear trend as in GCNNs. RotEqNet seem to be severely affected by FGSM even at no rotation augmentation. Their accuracy changes roughly from 40% to 60% as we augment the network from $\pm 0°$ to $\pm 180°$ range of random rotations.

## 3 DETAILS OF EXPERIMENTS

All experiments performed on neural network-based models were done using MNIST, Fashion MNIST and CIFAR10 datasets with appropriate augmentations applied to the train/validation/test set.

**Data sets**   MNIST[3] dataset consists of $70,000$ images of $28 \times 28$ size, divided into 10 classes. $55,000$ used for training, $5,000$ for validation and $10,000$ for testing. Fashion MNIST[4] dataset consists of $70,000$ images of $28 \times 28$ size, divided into 10 classes. $55,000$ used for training, $5,000$ for validation and $10,000$ for testing. CIFAR10[5] dataset consists of $60,000$ images of $32 \times 32$ size, divided into 10 classes. $40,000$ used for training, $10,000$ for validation and $10,000$ for testing.

**Model Architectures**   For the MNIST and Fashion MNIST based experiments we use the 7 layer architecture of GCNN similar to Cohen & Welling (2016). The StdCNN architecture is similar to the GCNN except that the operations are as per CNNs. Refer to Table 1 for details. RotEqNet architecture is as given in Marcos et al. (2017). For the CIFAR10 based experiments we use the ResNet18 architecture as in He et al. (2016) and its equivalent in GCNN as given in Cohen & Welling (2016). Input training data was augmented with random cropping and random horizontal flips by default for all experiments.

Table 1: Architectures used for the MNIST and Fashion MNIST experiments - 1

| Standard CNN | GCNN |
| --- | --- |
| Conv(10,3,3) + Relu | P4ConvZ2(10,3,3) + Relu |
| Conv(10,3,3) + Relu | P4ConvP4(10,3,3) + Relu |
| Max Pooling(2,2) | Group Spatial Max Pooling(2,2) |
| Conv(20,3,3) + Relu | P4ConvP4(20,3,3) + Relu |
| Conv(20,3,3) + Relu | P4ConvP4(20,3,3) + Relu |
| Max Pooling(2,2) | Group Spatial Max Pooling(2,2) |
| FC(50) + Relu | FC(50) + Relu |
| Dropout(0.5) | Dropout(0.5) |
| FC(10) + Softmax | FC(10) + Softmax |

---

[3]http://www.iro.umontreal.ca/ lisa/twiki/bin/view.cgi/Public/MnistVariations
[4]https://github.com/zalandoresearch/fashion-mnist
[5]https://www.cs.toronto.edu/ kriz/cifar.html

## 4 CONCLUSION

We observe that the robustness to geometric transformations in equivariant networks comes at the cost of their robustness to pixel-wise adversarial perturbations. We do an extensive comparative study of various equivariant network models ranging from StdCNNs to GCNNs, HNets, PTNs, RotEqNets for MNIST, Fashion MNIST and StdCNNs, GCNNs for CIFAR10 datasets. We believe that good neural network models should be robust to both geometric transformations and pixel-wise adversarial perturbations, and understanding trade-offs similar to the ones in our paper is an important direction for future work.

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

# A    APPENDIX

Table 2: Architectures used for the MNIST and Fashion MNIST experiments - 2

| Standard CNN | GCNN |
|---|---|
| Conv(20,3,3) + Relu | P4ConvZ2(5,3,3) + Relu |
| Conv(20,3,3) + Relu | P4ConvP4(5,3,3) + Relu |
| Max Pooling(2,2) | Group Spatial Max Pooling(2,2) |
| Conv(40,3,3) + Relu | P4ConvP4(10,3,3) + Relu |
| Conv(40,3,3) + Relu | P4ConvP4(10,3,3) + Relu |
| Max Pooling(2,2) | Group Spatial Max Pooling(2,2) |
| FC(50) + Relu | FC(50) + Relu |
| Dropout(0.5) | Dropout(0.5) |
| FC(10) + Softmax | FC(10) + Softmax |

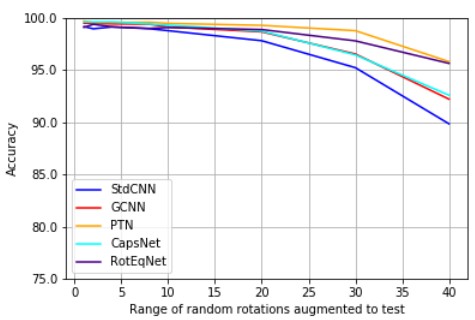

Figure 13: On MNIST, (left) networks trained with no augmentation, test augmented with random rotations in $[-x°, x°]$ range.

Apart from comparing StdCNN and GCNN with themselves that is when train has no rotation augmentation to train-augmented with rotation, we also compare the two networks by considering pixel-wise attack on architectures as given in Table 1 and 2. The networks in Table 2 roughly become parameter equivalent and channel equivalent for each layer in the architecture hence, allowing a fairer comparison between the two. We observe that as GCNNs become more robust to

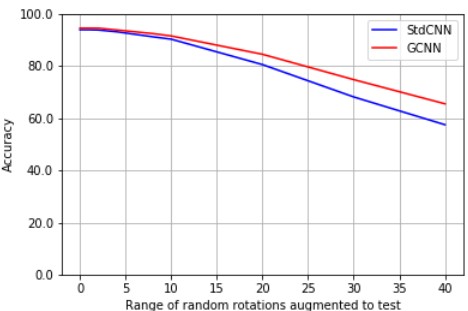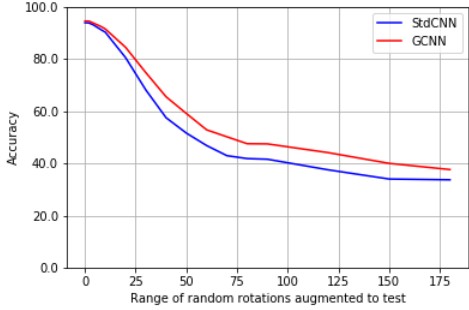

Figure 14: On CIFAR10, networks trained with no augmentation, test augmented with random rotations in $[-x°, x°]$ range. (left) Rotations(small) upto $40°$ (right) Rotations from $0°$ to $180°$

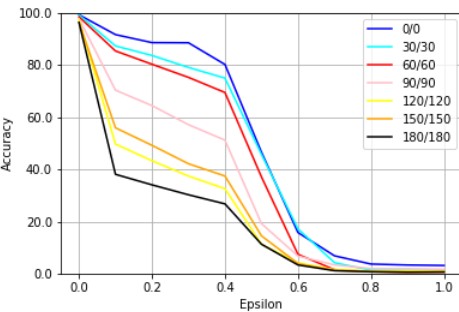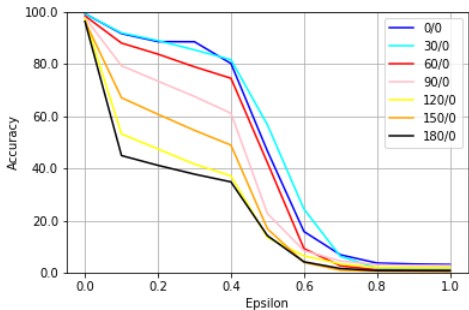

Figure 15: StdCNNs, MNIST adversarially attacked with PGD at varying epsilon ball as perturbation budget on X-axis. (left) train and test augmented with $[-x°, x°]$ range (right) Only train augmented with $[-x°, x°]$ range and no test augmentation.

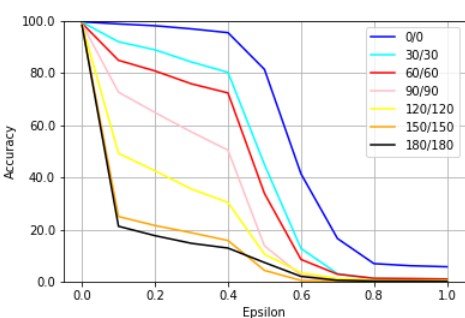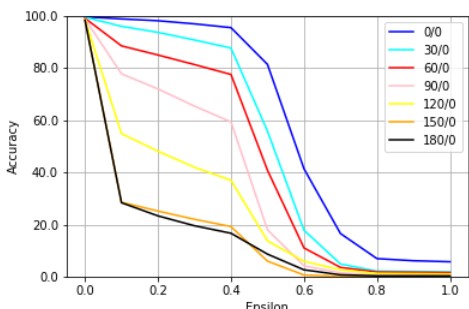

Figure 16: GCNNs, MNIST adversarially attacked with PGD at varying epsilon ball as perturbation budget on X-axis. (left) train and test augmented with $[-x°, x°]$ range (right) Only train augmented with $[-x°, x°]$ range and no test augmentation.

rotations they are more vulnerable to pixel-wise attacks, more so than StdCNNs. In parallel we also show what happens when the networks are adversarially trained and tested with/without adversarial perturbations. Please refer plots 22 to 28 for results on MNIST dataset. Refer to plots 29 to 32 for results on Fashion MNIST.

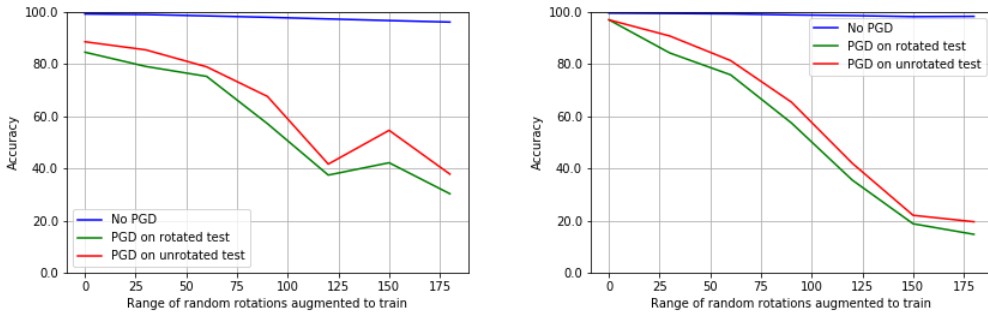

Figure 17: On MNIST, Comparison of network with/without PGD on rotated and unrotated test with $\epsilon = 0.3$. Train/test if augmented are with random rotations in $[-x°, x°]$ range (left) StdCNNs, (right) GCNNs.

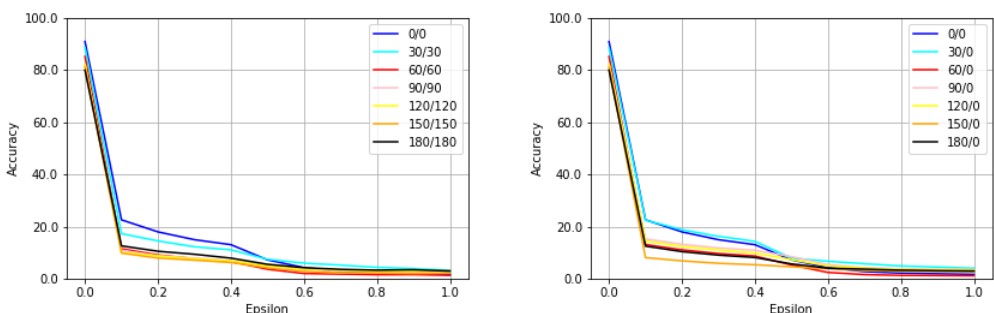

Figure 18: StdCNNs, Fashion MNIST adversarially attacked with PGD at varying epsilon ball as perturbation budget on X-axis. (left) train and test augmented with $[-x°, x°]$ range (right) Only train augmented with $[-x°, x°]$ range and no test augmentation.

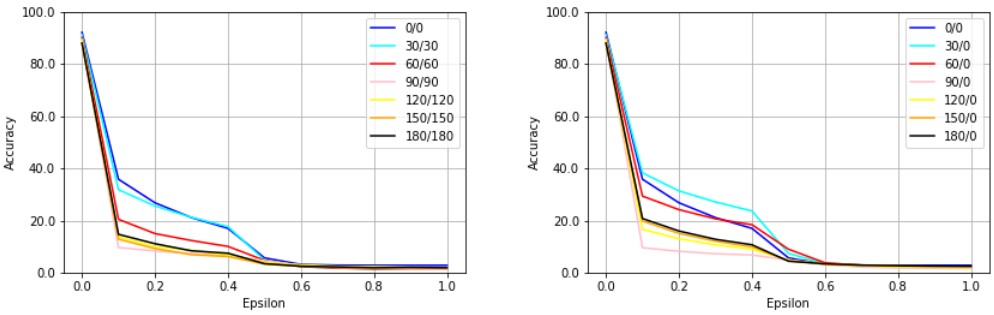

Figure 19: GCNNs, Fashion MNIST adversarially attacked with PGD at varying epsilon ball as perturbation budget on X-axis. (left) train and test augmented with $[-x°, x°]$ range (right) Only train augmented with $[-x°, x°]$ range and no test augmentation.

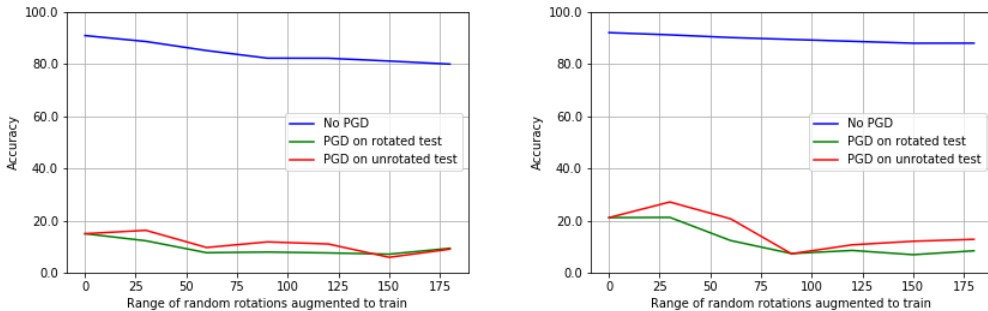

Figure 20: On Fashion MNIST, Comparison of network with/without PGD on rotated and unrotated test with $\epsilon = 0.3$. Train/test if augmented are with random rotations in $[-x^\circ, x^\circ]$ range. (left) StdCNNs, (right) GCNNs.

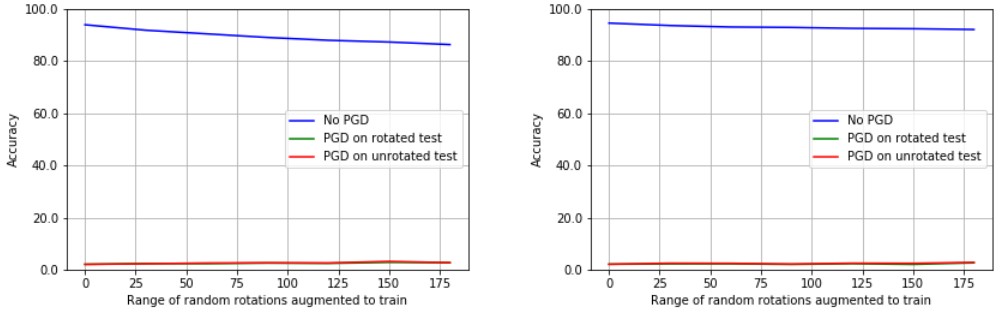

Figure 21: On CIFAR10, Comparison of network with/without PGD on rotated and unrotated test with $\epsilon = 0.06$. Train/test if augmented are with random rotations in $[-x^\circ, x^\circ]$ range. (left) StdCNNs, (right) GCNNs.

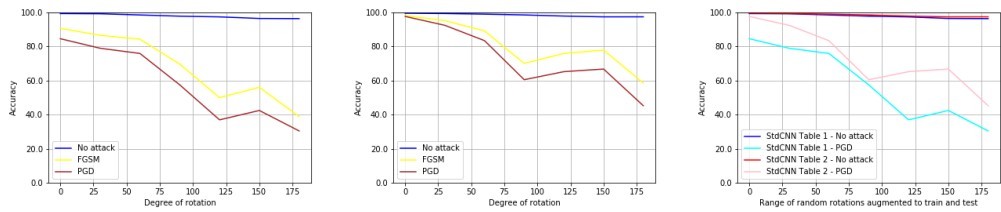

Figure 22: On MNIST, Comparison between Test with No attack, FGSM and PGD attack, $\epsilon = 0.3$. Train/test augmented with random rotations in $[-x^\circ, x^\circ]$ range. (left) StdCNN in table 1 (center) StdCNN in table 2, (right) PGD attack comparison between both StdCNN architectures from 1 and 2.

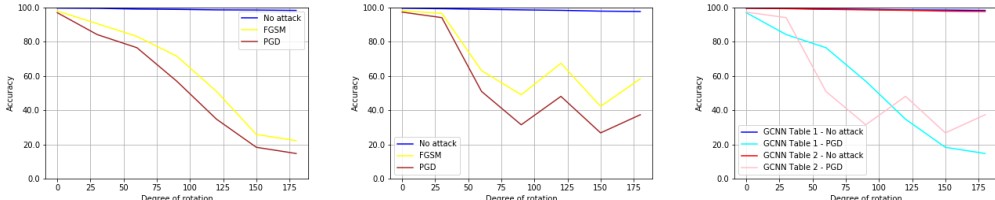

Figure 23: On MNIST, Comparison between Test with No attack, FGSM and PGD attack, $\epsilon = 0.3$. Train/test augmented with random rotations in $[-x°, x°]$ range. (left) GCNN in table 1 (center) GCNN in table 2, (right) PGD attack comparison between both GCNN architectures from 1 and 2.

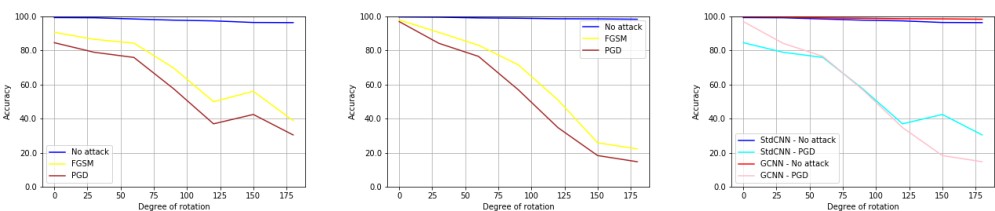

Figure 24: On MNIST, For Networks give in table 1, comparison between Test with No attack, FGSM and PGD attack, $\epsilon = 0.3$. Train/test augmented with random rotations in $[-x°, x°]$ range. (left) StdCNNs, (center) GCNNs, (right) StdCNNs vs GCNNs - PGD attack.

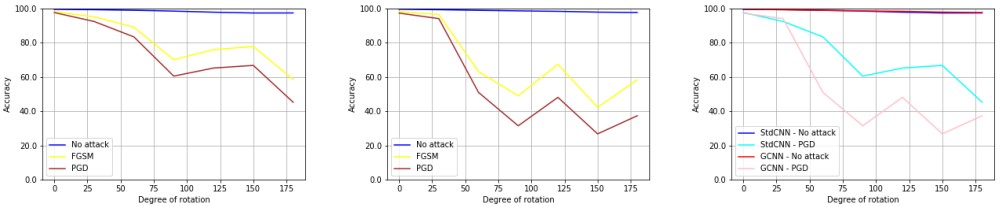

Figure 25: On MNIST, For Networks as given in table 2, comparison between Test with No attack, FGSM and PGD attack, $\epsilon = 0.3$. Train/test augmented with random rotations in $[-x°, x°]$ range. (left) StdCNNs, (center) GCNNs, (right) StdCNNs vs GCNNs - PGD attack.

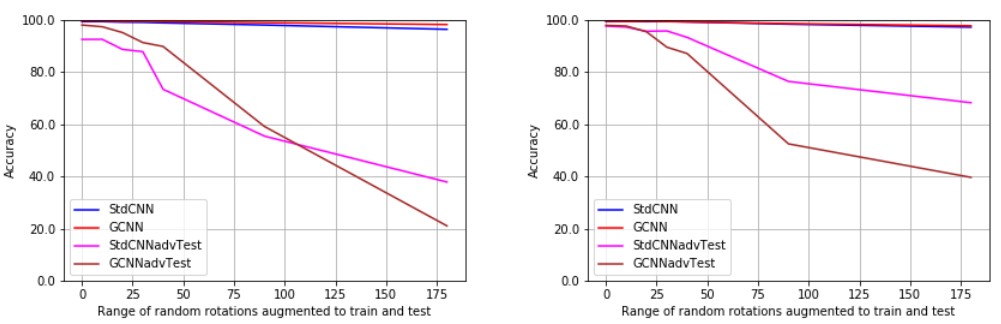

Figure 26: On MNIST, without FGSM training and test FGSM perturbed (left) Networks as given in Table1. Train/test augmented with random rotations in $[-x°, x°]$ range. (right) Networks as given in Table2. Train/test augmented with random rotations in $[-x°, x°]$ range.

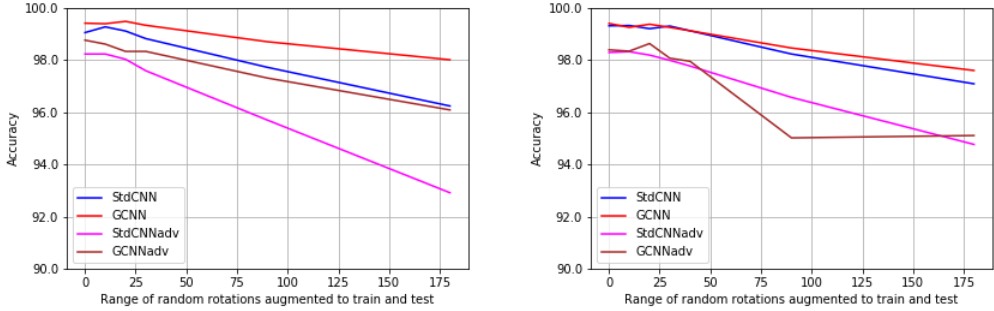

Figure 27: On MNIST, with both train and test FGSM perturbed (left) Networks as given in Table1. Train/test augmented with random rotations in $[-x°, x°]$ range. (right) Networks as given in Table2. Train/test augmented with random rotations in $[-x°, x°]$ range.

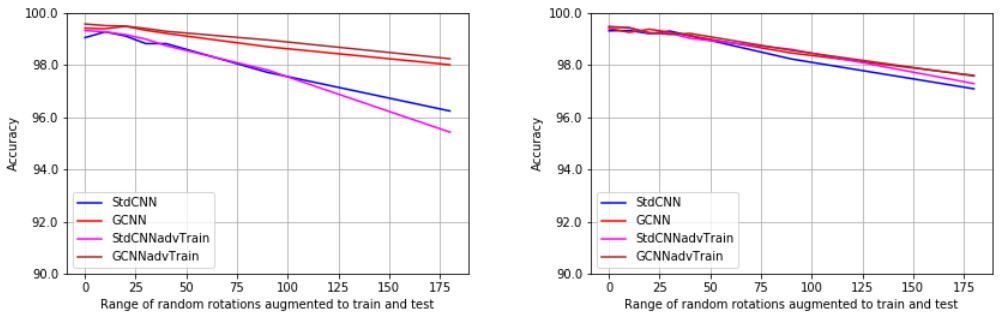

Figure 28: On MNIST, with FGSM adversarial training only (left) Networks as given in Table1. Train/test augmented with random rotations in $[-x°, x°]$ range. (right) Networks as given in Table2. Train/test augmented with random rotations in $[-x°, x°]$ range.

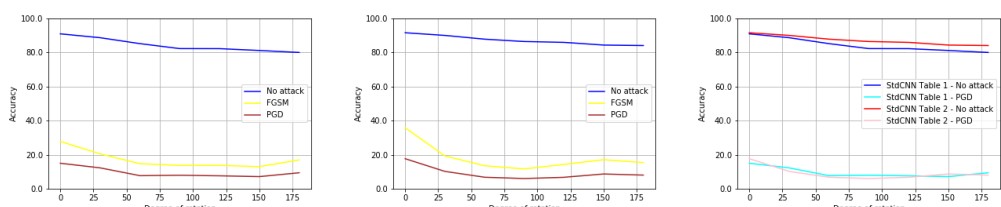

Figure 29: On Fashion MNIST, Comparison between Test with No attack, FGSM and PGD attack, $\epsilon = 0.3$. Train/test augmented with random rotations in $[-x°, x°]$ range. (left) StdCNN in table 1 (center) StdCNN in table 2, (right) PGD attack comparison between both StdCNN architectures from 1 and 2.

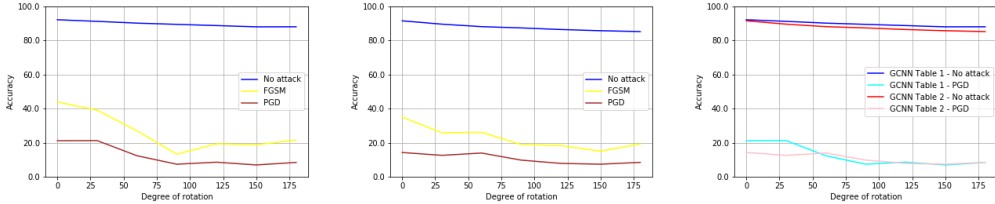

Figure 30: On Fashion MNIST, Comparison between Test with No attack, FGSM and PGD attack, $\epsilon = 0.3$. Train/test augmented with random rotations in $[-x°, x°]$ range. (left) GCNNs in table 1 (center) GCNNs table 2, (right) Both GCNN architectures from 1 and 2.

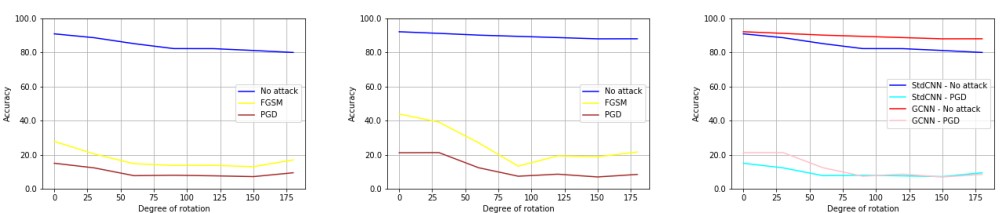

Figure 31: On Fashion MNIST, For Networks give in table 1, comparison between Test with No attack, FGSM and PGD attack, $\epsilon = 0.3$. Train/test augmented with random rotations in $[-x°, x°]$ range. (left) StdCNNs, (center) GCNNs, (right) StdCNNs vs GCNNs - PGD attack.

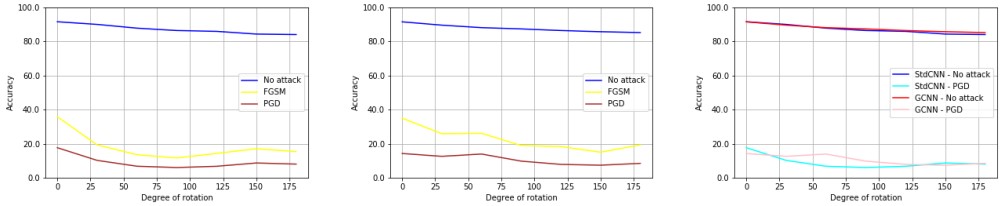

Figure 32: On Fashion MNIST, Comparison between Test with No attack, FGSM and PGD attack, $\epsilon = 0.3$. Train/test augmented with random rotations in $[-x°, x°]$ range. (left) GCNN in table 1 (center) GCNN in table 2, (right) PGD attack comparison between both GCNN architectures from 1 and 2.

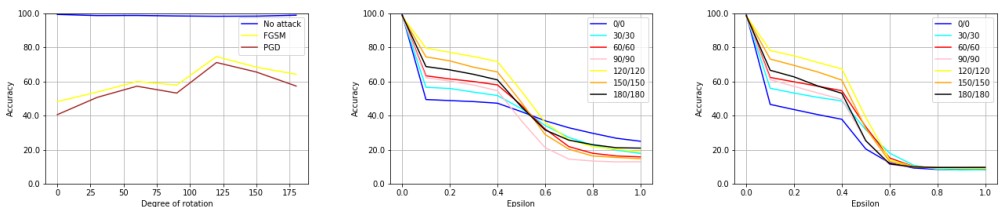

Figure 33: RotEqNet, MNIST with train and test augmented with $[-x°, x°]$ range, (left) Adversarial test with FGSM and PGD, $\epsilon = 0.3$, (center) Varying epsilon ball for adversarial test with FGSM, (right) Varying epsilon ball for adversarial test with PGD

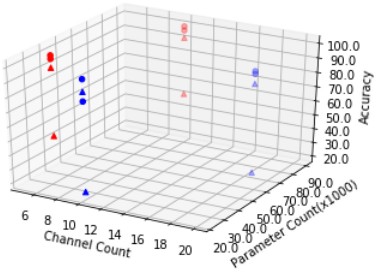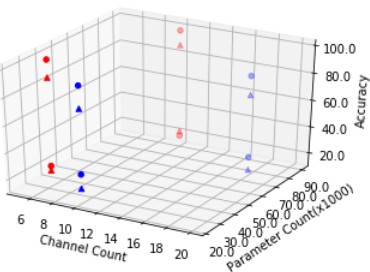

Figure 34: Scatter Plot, StdCNN(red) vs GCNN(blue), MNIST(circle) and Fashion MNIST(triangle) (left) No augmentation, (right) train and test augmented with $[-180°, 180°]$ range.

