# OpenReview forum: "Robustness and Equivariance of Neural Networks"
_ICLR.cc/2019/Conference_

### Official Review · AnonReviewer3 · 2018-10-22
**Robustness and Equivariance of Neural Networks**

**Rating:** 5
**Confidence:** 3

**Review:**

This paper empirically studies the robustness of equivariant CNNs to rotations as well as adversarial perturbations. It also studies their sample efficiency, parameter efficiency, and the effect of rotation- and adversarial augmentation during training and/or testing.

The main findings are:
1) Rotation-equivariant networks are robust to small rotations, even if equivariance to small rotations is not directly built into the architecture
2) Applying rotational data augmentation increases robustness to rotations
3) Equivariant networks are more sample efficient than CNNs and outperform them for all dataset sizes.
4) Applying rotational data augmentation decreases robustness to adversarial perturbations, and this effect is more pronounced for GCNNs.

If true, this is a valuable addition to the literature. It is one of the first independent validations of claims regarding sample complexity and accuracy made by the authors of the various equivariant network papers, performed by a party that does not have their own method to promote. Many of the findings do not have an obvious explanation, so the data from this paper could conceivably prompt new theoretical questions and investigations.

The authors chose to highlight one finding in particular, namely that GCNNs become more sensitive to adversarial perturbations as they are trained on more heavily rotation-augmented data. However, this appears to be true for both CNNs and GCNNs, the difference being only in degree (see fig 4, 10, 11). This is not apparent from the text though, as e.g. the abstract notes that "robustness to geometric transformations in these models [equivariant nets] comes at the cost of robustness to small pixel-wise perturbations".

Since HNets, GCNNs and RotEqNets should be exactly equivariant to 90 degree rotations (and some others, perhaps), it is surprising that figure 1 shows a continuing decline in performance with bigger and bigger random rotations. If the network is made rotation invariant through some pooling layer at the end of the network, one would expect to see a decline in performance up to 45 degrees, followed by an increase back to baseline at 90 degrees, etc.

Polar transformer networks achieve good results in fig. 1, but I wonder if this is still true for rotations around points other than the origin.

Since CNNs and GCNNs differ in terms of the number of channels at a certain number of parameters, and differ in terms of number of parameters at a certain number of channels, it could be that channel count or parameter count is the more relevant factor, rather than equivariance. So it would be good to make a scatterplot where each dot is a network (either CNN or GCNN, at various model sizes), the x-axis is parameter count (or in another plot, 2d channel count), and the y-axis corresponds to the accuracy. This can be done for various choices of augmentation / perturbation. The type of network (CNN or GCNN) could be color coded. If indeed the CNN/GCNN variable is relevant, that should be clearly visible in the plot, and similarly if the parameter count or channel count is relevant. One could also do a linear regression of accuracy or log-accuracy or something using CNN/GCCN, param-count, channel-count as covariates, and report the variance explained by each.

In several plots, e.g. fig 4, 8, the y-axes do not have the same range, making it hard to compare results between subplots.

The experiments have some weaknesses. For one thing, it seems like each accuracy value reported comes from a single training run. It would be much preferable to plot mean and standard deviation / error bars. Another weakness is that all experiments are performed on MNIST. Even just a simple validation of the main findings on CIFAR would significantly strengthen the paper.

Because of the limited scope of the experiments, it is not clear to me how generalizable and robust the experimental results are. With deep network performance it can be hard to know what the relevant hyperparameters are, and so we may well be reading tea leaves here.

It is also unfortunate that no explanation for the observed phenomena is available. However, it is conceivable that the findings presented in this paper could help researchers who are trying to understand adversarial attacks / robustness, so it is not a fatal flaw. I am certainly glad the authors did not make up some unsupported story to explain the findings, as is all too common in the literature these days.

Overall, I consider this a borderline paper, and am tending towards a reject. My main considerations are:
1. Uncertainty about generalizability
2. Uncertainty about usefulness to practitioners or theorists (admittedly, this is hard to predict, but no clear use-case is available at this point)
3. A lot of data, but no clear central finding of the paper

---

> ### Author Response · Authors · 2018-11-26
> **Central finding explained, and revised version with CIFAR10 and Fashion MNIST experiments**
>
> The central finding of our paper is a novel empirical observation of the following trade-off: as we train StdCNNs and GCNNs with rotation augmentations to make them more robust to rotations, they instead become more vulnerable to pixel-wise adversarial attacks. This contrasts starkly with previous work that mostly treats the robustness to geometric transformations and the robustness to pixel-wise adversarial attacks as somewhat independent of each other.
>
> We also do a comprehensive comparison of the effectiveness of various models in learning rotations, starting from StdCNNs to rotation-equivariant models such as GCNNs, HNets, PTNs, RotEqNets etc. We have revised our submission with experiments on Fashion MNIST and CIFAR10 that validate our central finding. Please see Figures 2, 3, 4, 9-12.
>
> Below is our detailed pointwise response to various important concerns raised in the review.
>
> -- We can only conclude that both CNNs and GCNNs become more vulnerable to pixel-wise adversarial perturbations as they are train-augmented with a larger range of rotations. We do not mean any CNNs-vs-GCNNs comparison in the statement "robustness to geometric transformations in these models [equivariant nets] comes at the cost of robustness to small pixel-wise perturbations".
>
> -- In Fig. 1, the X-axis represents test augmentation with random rotations in the range [-x degrees, +x degrees]. At 90 degrees, the plot represents test accuracy with test input from MNIST augmented with random rotations in the entire range between -90 degrees and +90 degrees. Therefore, even though HNets, GCNNs and RotEqNets are equivariant to exactly 90 degree rotations, the test accuracy declines.
>
> -- In Fig. 2, we have an example using Fashion MNIST where Polar Transformer Networks (PTNs) work poorly even for rotation around the center (compared to their performance on MNIST). We believe this is due to the paucity of natural rotation augmentations in Fashion MNIST (compared to MNIST).
>
> -- As per your suggestion, we have included a scatter plot (see Fig. 34 in Appendix) using X-axis for the channel count, Y-axis for the parameter count and Z-axis for the accuracy, with color-coding for the type of network (CNN or GCNN) and shape-coding for the data set (MNIST, Fashion MNIST). We can add more experiments and the regression plot in later revisions, if possible.
>
> -- We have revised the Y-axes of subplots to have the same range. Thank you for the valuable suggestion to improve our presentation.
>
> -- We have included experiments on Fashion MNIST and CIFAR10, and revised the submission.
>
> -- Although we have not plotted the mean and standard deviation/error bars, we have noticed the rough error bars within +/-1% on MNIST and +/-2% on CIFAR10 for our experiments with StdCNNs and GCNNs over multiple runs. We can include these, if that helps.

---

### Official Review · AnonReviewer1 · 2018-10-29
**Not good enough to accept**

**Rating:** 4
**Confidence:** 4

**Review:**

Using the dataset MNIST, the authors empirically studied the robustness of several rotation-equivariant neural network models(GCNN, H-Nets, PTN, et al.) to geometric transformation and small pixel-wise perturbations. Their experiments showed that the equivariant network models(StdCNNs, GCNNs, H-Nets, et al.) are robust to geometric transformation but vulnerable to pixel-wise adversarial perturbations. These findings help us understand the  neural network models better.
However, this paper is not acceptable due to lack of innovation and novelty.

---

> ### Author Response · Authors · 2018-11-26
> **Revised version with CIFAR10 and Fashion MNIST experiments**
>
> Our novel empirical observation is the following trade-off: as we train StdCNNs and GCNNs with rotation augmentations to make them more robust to rotations, they instead become more vulnerable to pixel-wise adversarial attacks. This contrasts starkly with previous work that mostly treats the robustness to geometric transformations and the robustness to pixel-wise adversarial attacks as somewhat independent of each other.
>
> We also do a comprehensive comparison of the effectiveness of various models in learning rotations, starting from StdCNNs to rotation-equivariant models such as GCNNs, HNets, PTNs, RotEqNets etc. We have revised our submission with experiments on Fashion MNIST and CIFAR10, as you had asked in the review. Please see Figures 2, 3, 4, 9-12.

---

### Official Review · AnonReviewer2 · 2018-10-31
**Interesting empirical study of CNN robustness, potentially good workshop paper**

**Rating:** 3
**Confidence:** 5

**Review:**

This paper empirically studies various CNN robustifying mechanisms aiming to achieve rotational invariance. The main finding is that such robustifying mechanisms may lead to lack of robustness against pixel-level attacks such as FGSM and its variants. The paper does a comprehensive job in studying relevant robustifying schemes and attacks strategies. However, the paper does not present sufficiently new information worthy of a regular conference paper, it can be a good workshop paper though for the Robust Learning community. Some analytical insights would really strengthen the work. Also, from an empirical standpoint, the authors need to consider other data sets beyond just the MNIST data set.

xxxxxxxxxxxxxx

While I appreciate the authors' rebuttal and revisions, I still do not see sufficient contribution here worthy of a regular ICLR paper.

---

> ### Author Response · Authors · 2018-11-26
> **Revised version with CIFAR10 and Fashion MNIST experiments**
>
> Our novel empirical observation is the following trade-off: as we train StdCNNs and GCNNs with rotation augmentations to make them more robust to rotations, they instead become more vulnerable to pixel-wise adversarial attacks. This contrasts starkly with previous work that mostly treats the robustness to geometric transformations and the robustness to pixel-wise adversarial attacks as somewhat independent of each other.
>
> We also do a comprehensive comparison of the effectiveness of various models in learning rotations, starting from StdCNNs to rotation-equivariant models such as GCNNs, HNets, PTNs, RotEqNets etc. We have revised our submission with experiments on Fashion MNIST and CIFAR10, as you had asked in the review. Please see Figures 2, 3, 4, 9-12.

---

### Meta-Review · Area_Chair1 · 2018-12-13
**Area chair recommendation**

**Confidence:** 5
**Recommendation:** Reject

**Metareview:**

Positives:

The paper proposes an interesting idea: to study the effect on vulnerability to adversarial attacks of training for invariance with respect to rotations.
Experiments on MNIST, FashionMNIST, and CIFAR10.
An interesting hypothesis partially borne out in experiments.

Negatives:

no accept recommendation from any reviewer
insufficient empirical results
not a clear enough message
very limited theoretical contribution

Although additional experimental results on FashionMNIST and CIFAR10 were added to the initial very limited results on MNIST, the main claim of the paper seems to be somewhat weakened.  The effect of increased vulnerability to adversarial attacks as invariance is increased is less pronounced on the additional datasets.  This calls into question how relevant this effect is on more realistic data than the toy problems considered here.

The size of the network is not varied in the experiments.  If increased invariance results in poorer performance with respect to attacks, one possible explanation is that the invariance taxes the capacity of the network architecture.  Varying architecture depth could partially answer whether this is relevant.  Given the lack of theoretical contribution, more insights along these lines would potentially strengthen the work.

The title uses the term "equivariance," which strictly speaking is when the inputs and outputs of a function vary equally, e.g. an image and its segmentation are equivariant under rotations, but classification tasks should probably be called "invariant."

The reviewers were unanimous in not recommending the paper for acceptance.  The key concerns remain after the author response.